# Efficacy of Early Optimization of Infliximab Guided by Therapeutic Drug Monitoring during Induction—A Prospective Trial

**DOI:** 10.3390/biomedicines11061757

**Published:** 2023-06-19

**Authors:** Karoline Soares Garcia, Matheus Freitas Cardoso de Azevedo, Alexandre de Sousa Carlos, Luísa Leite Barros, Jane Oba, Carlos Walter Sobrado Junior, Aytan Miranda Sipahi, Olívia Duarte de Castro Alves, Tomás Navarro-Rodriguez, Rogério Serafim Parra, Júlio Maria Fonseca Chebli, Liliana Andrade Chebli, Cristina Flores, Andrea Vieira, Christianne Damasceno Arcelino do Ceará, Natália Sousa Freitas Queiroz, Aderson Omar Mourão Cintra Damião

**Affiliations:** 1Department of Gastroenterology, University of São Paulo, School of Medicine, São Paulo 05403-000, Brazil; 2Ribeirão Preto Medical School, University of São Paulo, Ribeirão Preto 14049-900, Brazil; 3University Hospital of the Federal University of Juiz de Fora, Juiz de Fora 36038-330, Brazil; 4Crohn’s and Colitis Reference Center, Rio Grande do Sul 90560-002, Brazil; 5Irmandade da Santa Casa de Misericórdia de São Paulo, São Paulo 01221-010, Brazil; 6Health Sciences Graduate Program, Pontifícia Universidade Católica do Paraná (PUCPR), Curitiba 80215-901, Brazil; 7IBD Center, Santa Cruz Hospital, Curitiba 80420-090, Brazil

**Keywords:** inflammatory bowel diseases, tumor necrosis factor-alpha, infliximab, therapeutic drug monitoring, Crohn’s disease, ulcerative colitis

## Abstract

Therapeutic drug monitoring (TDM) during induction therapy with anti-tumor necrosis factor drugs has emerged as a strategy to optimize response to these biologics and avoid undesired outcomes related to inadequate drug exposure. This study aimed to describe clinical, biological, and endoscopic remission rates at six months in Brazilian inflammatory bowel disease (IBD) patients following a proactive TDM algorithm guided by IFX trough levels (ITL) and antibodies to IFX (ATI) levels during induction, at week six. A total of 111 IBD patients were prospectively enrolled, excluding those previously exposed to the drug. ITL ≥ 10 μg/mL was considered optimal. Patients with suboptimal ITL (<10 µg/mL) were guided according to ATI levels. Those who presented ATI ≤ 200 ng/mL underwent dose intensification in the maintenance phase, and patients with ATI > 200 ng/mL discontinued IFX. In our study, proactive TDM was associated with persistence in the IFX rate at six months of 82.9%. At that time, rates of clinical, biological, and endoscopic remission in patients under IFX treatment were 80.2%, 73.9%, and 48.1%, respectively. Applying a simplified TDM-guided algorithm during induction seems feasible and can help improve patients’ outcomes in clinical practice.

## 1. Introduction

Despite the development and approval of new drugs targeting different pathways involved in inflammatory bowel disease (IBD), infliximab (IFX) remains a highly effective treatment for inducing and maintaining remission in Crohn’s disease (CD) and ulcerative colitis (UC) [1,2,3]. However, up to 40% of patients may present a primary non-response (PNR) to infliximab [4,5,6]. Furthermore, approximately 40% of responders experience clinical remission, and only 30% show endoscopic remission at six months [4,5]. Over one year, about 50% of patients discontinue treatment due to loss of response or adverse effects [4,5]. The mechanisms underlying these undesired outcomes have been associated with pharmacokinetics—inadequate drug exposure, either due to low drug level or anti-drug antibodies (ADA)—and pharmacodynamics—inflammatory process unrelated to the targeted immunoinflammatory pathway [5,6,7].

The development of drug assays for biological drugs has led to studies showing a positive association between serum drug concentrations and favorable therapeutic outcomes have been published [8,9,10]. These findings have supported the utility of assessing drugs and antibodies to the drug levels to guide therapeutic decisions, termed therapeutic drug monitoring (TDM). Unlike reactive TDM, defined by evaluating these laboratory parameters to guide interventions just after the loss of response, proactive therapeutic drug monitoring (pTDM) is performed to guide dose intensification in patients responding to treatment to prevent loss of response. Thus, pTDM has emerged as a strategy to optimize biological therapy and maximize its effectiveness, improving persistence in therapy and reducing the risk of treatment failure (drug discontinuation or need for surgery) as well as the risk of hospitalization, development of ADA, and infusion reactions [11]. In addition, preliminary data from observational studies demonstrate that pTDM (with drug titration to a target trough concentration) during induction is associated with better therapeutic outcomes than empiric dose optimization [12,13].

This study aimed to evaluate 6-month clinical, biological, and endoscopic remission rates in a cohort of IBD patients following a TDM-guided algorithm during infliximab induction. We hypothesized that dosing decisions based on infliximab trough level (ITL) and antibodies to infliximab (ATI) at week six effectively improve clinical, biological, and endoscopic remission rates and enhance IFX durability. 

## 2. Materials and Methods

### 2.1. Study Design and Patient Population

This multicenter, prospective, interventional, open-label study was conducted at five IBD referral centers in Brazil’s South and Southeast regions from December 2021 to August 2022. Inclusion criteria were IBD outpatients aged 18 years or older who would receive infliximab (Remicade^®^, Janssen: Zurich, Switzerland) at a standard induction dose (5 mg/kg by intravenous infusion at 0, 2, and 6 weeks). No patient received the biosimilar of IFX. Enrolled patients followed an algorithm that proactively adjusted IFX dosage for the maintenance phase according to ITL and ATI levels measured before the week-six infusion (Figure 1). The algorithm was proposed by Sparrow et al. [11] and Papamichael et al. [14], and it resulted from studies supporting higher anti-TNF drug levels during induction were associated with favorable outcomes in short and long terms in IBD patients initiating treatment with infliximab, adalimumab, or certolizumab pegol. Early optimization of anti-TNF therapy based on this approach might also prevent undesired outcomes, such as primary non-response and the need for surgeries and hospitalizations, especially when there is an increased risk of higher drug clearance [11,14]. ITL greater than or equal to 10 µg/mL at week six was considered optimal; patients in this scenario were referred to receive infliximab during the maintenance phase at standard doses. At week six, patients with a suboptimal ITL (less than 10 µg/mL) were also evaluated according to ATI level. Patients with low to moderate ATI (less than or equal to 200 ng/mL) underwent infliximab dose intensification in the maintenance phase, with the addition of an immunomodulator if it was not already being used. Intensified IFX treatment consisted of IFX at 10 mg/kg every eight weeks, 5 mg/kg every four weeks, or 5 mg/kg every six weeks, according to physician judgment. Patients with high ATI titer (greater than 200 ng/mL) switched therapy and were excluded from the study (Figure 2). Patients were followed for six months or until infliximab discontinuation if earlier than six months. Patients previously exposed to IFX, with incomplete therapeutic information, or without follow-up were excluded.

### 2.2. Infliximab Therapeutic Drug Monitoring

ITL was determined from serum samples collected 0 to 2 days before infliximab infusion at week six (induction phase). ATI was also evaluated if ITL was less than 10 µg/mL in the sample. Serum ITL and ATI were determined using a commercially available and validated enzyme-linked immunosorbent assay (ELISA) kit (Lisa Tracker^®^, Theradiag: Croissy-Beaubourg, France). The kit is based on a two-step test with biotinylated anti-human IgG antibodies and horseradish peroxidase streptavidin. The limits of quantification for infliximab were from 0.3 to 20 µg/mL, and for ATI, they were from 10 to 200 ng/mL. 

### 2.3. Data Collection

The following baseline variables were collected: gender, age, body mass index (BMI), smoking, type of IBD, age at IBD diagnosis, duration of IBD, the extent of disease and behavior according to the Montreal classification, history of previous use of biologics or oral small molecules, concomitant use of immunomodulators and corticosteroids, and serum level of C-reactive protein (CRP). Data on disease activity indexes (partial Mayo score (PMS) for UC and Harvey–Bradshaw index (HBI) for CD) and endoscopic activity indexes (Mayo endoscopic score (MES) for UC, simple endoscopic score for CD (SES-CD) and Rutgeerts score for CD patients with ileocolonic resection) were also recorded. 

### 2.4. Outcomes Measures

Primary outcomes were clinical, biological, and endoscopic remission rates at six months. Clinical remission was defined as HBI < 5 for CD patients or PMS ≤ 2, with all individual categories ≤ 1, for UC patients. Biological remission was defined as CRP within normal limits (≤5 mg/L). Endoscopic remission was defined by MES ≤ 1 for UC, SES-CD ≤ 2 for CD, or a Rutgeerts score of i0 or i1 for post-operative CD. The secondary outcome included IFX retention rates at six months.

### 2.5. Statistical Analysis

Statistical analysis was performed using the IBM-SPSS for Windows version 22.0 software (IBM Corp., Armonk, NY, USA) and tabulated using the Microsoft-Excel 2013 software (Microsoft, Redmond, WA, USA).

The qualitative variables evaluated were described using absolute and relative frequencies, and the quantitative characteristics were described using summary measures (median, interquartile range). For statistical analysis, laboratory measurements with values below and above the detection limit were considered as the limit itself. Thus, for the CRP items, values below five mg/L were considered 5; while infliximab serum levels below 0.3 µg/mL were considered 0.3, and a value greater than 20 mg/L was considered 20.

The frequencies of the outcomes of interest (clinical, biological, and endoscopic remission) were described according to the pre-treatment qualitative characteristics using absolute and relative frequencies and the associations with the outcomes using the chi-square test or exact tests (Fisher’s exact test or likelihood ratio test). Quantitative characteristics were described according to results using summary measures and compared employing the Student’s *t*-test or Mann–Whitney test. The odds ratios (OR) were calculated with the respective unadjusted 95% confidence intervals for each characteristic evaluated for each outcome using bivariate logistic regression. The multiple logistic regression models were adjusted, inserting the features that presented descriptive levels in the unadjusted analysis values of less than 0.1 (*p* < 0.10) for outcomes in which the sample was sufficient to provide a multiple analysis, with all variables initially inserted being maintained in the final models (full model). The tests were performed with a significance level of 5%.

### 2.6. Ethical Considerations

All procedures followed the ethical standards of the Responsible Committee on Human Experimentation and the Declaration of Helsinki. All patients provided signed informed consent per local, regulatory, and legal requirements before initiating the study-related procedures. The local Ethical Research Committee approved the study under the number 51767421.0.0000.0068.

## 3. Results

### 3.1. Study Population Characteristics

The demographic and clinical characteristics of the 111 patients studied are presented in Table 1. Overall, 65 (58.6%) were female, and the median age was 37 (IQR 25–50). Most patients had CD (76 patients; 68.5%), while 35 (31.5%) had UC. Seventy-three patients (65.8%) took concomitant immunomodulators, with azathioprine being the most frequent (66 patients; 59.5%). Eighty-seven patients (78.4%) were not previously exposed to a biological drug (bio naïve). In contrast, 16 (14.4%) were exposed to adalimumab, one (0.9%) to certolizumab pegol, one (0.9%) to ustekinumab, two (1.8%) to etrolizumab, one (0.9%) to vedolizumab, and three (2.7%) to adalimumab and certolizumab pegol. Regarding oral small molecules, one patient (0.9%) was previously exposed to tofacitinib, and another (0.9%) to upadacitinib. 

### 3.2. Post-Induction Strategy According to the Algorithm

Among the 70 patients with ITL ≥ 10 µg/mL (group 1), two discontinued the drug in the induction phase—one patient due to PNR and the other due to tuberculosis infection. Finally, 68 patients proceeded to the maintenance phase with IFX at the standard dose (5 mg/kg every eight weeks). In the maintenance phase, two patients discontinued treatment due to loss of response, and two were excluded from the study because they missed some drug infusions. In total, 64 patients (91.4%) in this group persisted using IFX for six months (Figure 3). 

Of the 33 patients with ITL < 10 µg/mL but undetectable ATI (group 2), two discontinued the drug in the induction phase—one patient due to PNR and the other owing to a severe infusion reaction. Thirty-one patients underwent the maintenance phase with intensified IFX therapy—24 with IFX 10 mg/kg every eight weeks, one with IFX 5 mg/kg every six weeks, and six with IFX 5 mg/kg every four weeks. However, despite the drug optimization, five patients discontinued the drug during the maintenance phase due to loss of response. The rate of drug persistence in this group was 78.8 at six months (Figure 3). 

Among the eight patients who presented ITL < 10 µg/mL, with detectable ATI (group 3), three switched the drug due to a high titer of ATI (greater than 200 ng/mL). Three patients discontinued the medication in the induction phase—one because of PNR and two due to a severe infusion reaction. Finally, two patients (25%) continued for the maintenance phase with intensified IFX therapy (10 mg/kg every eight weeks) and were followed up for six months (Figure 3).

### 3.3. Clinical Remission at Six Months

Of the 111 patients who started IFX therapy, 92 (82.9%) persisted in being treated with IFX for six months. Most patients (73/91; 80.2%) achieved clinical remission; one had an ostomy and could not be evaluated according to the HBI (Figure 4). 

### 3.4. Biological Remission at Six Months

Biological remission was achieved in 68/92 subjects (73.9%)—Figure 4. Figure 5 demonstrates that biological remission rates were significantly higher at six months than pre-treatment CRP levels.

### 3.5. Endoscopic Remission at Six Months

Among 111 patients who started IFX, 38/79 (48.1%) patients achieved endoscopic remission at six months (Figure 4). Two patients did not undergo the procedure, and 11 had a disease that could not be evaluated by ileocolonoscopy. 

### 3.6. Predictors of Outcomes

The median ITL at week six was 14.5 µg/mL (IQR 5.7–20). Seventy patients (63.1%) had ITL ≥ 10 µg/mL, while 41 (36.9%) had ITL < 10 µg/mL. Among patients with suboptimal ITL, 8/41 (19.5%) presented detectable ATI, and three had ATI > 200 ng/mL. 

Figure 6 shows that the chi-square test for trend did not show a linear association between the quartiles of serum IFX level at week 6 with clinical or endoscopic remission at six months (*p* > 0.05).

Analyses per population are shown in Figure 7. Table 2 and Table 3 show that none of the evaluated characteristics showed a statistically significant relationship with clinical and endoscopic remission rates, respectively, when considered separately (*p* > 0.05). Table 4 shows that biological remission at six months in patients with UC was statistically higher than in those with CD (*p* = 0.021); also, the baseline CRP of patients in biological remission at six months was statistically lower (*p* = 0.002). Nevertheless, when evaluated in logistic regression, clinical remission was statistically influenced by the previous use of adalimumab, regardless of the other assessed characteristics (*p* = 0.045); patients who were previously exposed to adalimumab had 76% less chance of clinical remission than those who did not take this drug (Table 5).

## 4. Discussion

This study reports the characteristics and outcomes of 111 IBD patients following a simplified TDM-guided algorithm during induction treatment with IFX. Clinical, biological, and endoscopic remission rates at six months were 80.2%, 73.9%, and 48.1%, respectively. The drug persistence rate in this cohort was 82.9%. The previous exposure to adalimumab was significantly associated with clinical remission (*p* = 0.045). No other baseline characteristic was associated with the outcomes of interest.

Many reasons have led to the development of strategies to optimize the use of biological therapies in IBD. First, reaching relevant targets is imperative, as proposed in STRIDE-II [15]. Additionally, improving rates of drug persistence is of utmost importance since data have demonstrated that patients who fail anti-tumor necrosis factor (anti-TNF) therapies usually do not respond well to subsequent agents [10]. One such strategy is using immunomodulators in combination with anti-TNF, particularly with infliximab [16]. However, more recent studies have focused on measuring and intensifying doses of biologics early whenever the serum drug level is low, provided antibodies to the drug are not at high titers [17,18]. Despite the promising results of this so-called proactive approach, its application at the induction phase is still debatable, and pTDM in this specific phase is not already endorsed by gastroenterological societies [19,20].

The results of this study suggest that patients on infliximab who were managed based on pTDM at week six had higher rates of clinical remission (80.2% in total; 79.1% for CD; 83.3% for UC) and endoscopic remission (48.1% in total; 43.6% for CD; 58.3% for UC) at six months when compared with remission rates previously reported in the literature in patients following the conventional therapeutic approach. In ACCENT I, clinical remission was achieved in 39% of CD patients at week 30 [1]. In ACT-2 (active ulcerative colitis pivotal trial), clinical remission was achieved in 47.1% of UC patients [3]. This difference may be driven mainly by the differences in study design (randomized controlled trial x open-label), which prevents direct comparisons. Only a few studies reported remission rates in patients following a pTDM strategy. In a survey by Bossuyt et al. that aimed to compare the clinical outcomes of an ultra-proactive TDM algorithm of IFX based on point-of-care testing with reactive TDM, the rates of sustained clinical remission were 75% in the ultra-proactive versus 83% in the reactive group (*p* = 0.17). However, the study was not equipped to explore the potential benefits of pTDM during induction, given that all patients in the trial were in a maintenance treatment regimen at the time of inclusion [21]. 

The induction phase might be ideal for measuring serum drug concentrations since decisions could be made early. Another essential point associated with this period is that the inflammatory burden of active disease results in higher drug clearance and lower serum drug concentrations, which can favor the development of antidrug antibodies [14,22,23]. Given this, many observational studies have explored the relationship between infliximab trough concentrations at induction and desired outcomes [8,24,25,26,27], as well as the relationship between inadequate serum drug levels and high titers of antibodies to infliximab and PNR [11,28,29]. For example, Papamichael et al. found that ATI at week six was associated with a lack of mucosal healing after induction therapy in UC patients [24]. The authors also showed that ITL greater than or equal to 15 µg/mL at week six was associated with short-term mucosal healing, assessed between weeks 10 and 14 [24]. Similarly, a post hoc analysis of 484 UC patients from ACT 1/2 trials demonstrated that ITL greater than or equal to 18.6 µg/mL at week two and 10.6 µg/mL at week six were associated with endoscopic healing at week eight [25]. Additionally, Davidov et al. found that an ITL cutoff of 9.3 µg/mL at week two was a good predictor of fistula response at week 14 [26]. Conversely, the NOR-DRUM (Norwegian Drug Monitoring) trial, a randomized, multicenter, open-label study, evaluated 411 patients with chronic immune-mediated inflammatory diseases, including 147 IBD patients, who were initiated on infliximab therapy. Compared to those receiving clinically based dosing, no significant difference in clinical remission rates at week 30 was observed among subjects undergoing pTDM during induction. Notably, the trial had insufficient statistical power to test hypotheses within the IBD subgroup. Furthermore, in NOR-DRUM, the standard of care allowed for liberal dose increases in infliximab according to physicians’ judgments. This may have allowed this group attains a high efficacy rate, minimizing differences from the pTDM group [30]. 

Several potential explanations have been raised to underscore the performance of pTDM during induction as compared to standard therapy. For instance, it has been speculated that pTDM may be beneficial only in some subgroups of patients, such as those with perianal disease, patients previously exposed to other anti-TNF, and those with more severe disease [19]. Such cases were predominant in the population evaluated in the current study. Another population that could benefit from early pTDM is that being treated with IFX monotherapy, which differed from most patients in our cohort. Without a concomitant immunomodulator, pTDM may improve IFX durability by maintaining higher IFX levels [19,31]. In a post hoc analysis of 206 CD patients treated with IFX (monotherapy or in combination with azathioprine) from the SONIC (Study of Biologic and Immunomodulator-Naive Patients in Crohn’s Disease) trial, individuals presenting the same quartiles of ITL at week 30 achieved similar rates of corticosteroid-free remission and mucosal healing at week 26, suggesting that if adequate ITL is attained, combination therapy with thiopurine may not be required to achieve desired therapeutic outcomes [16]. 

In this study, we designed a simplified algorithm for pTDM during infliximab induction based on those proposed by Papamichael et al. and Sparrow et al. [11,14]. Although the threshold drug trough levels and the best timepoint of assessment have not yet been convincingly established, they proposed that in the presence of an adequate ITL at week six, there is no recommendation for measuring ATI since, in this scenario, anti-drug antibodies are unlikely to be clinically relevant. Patients achieving the target ITL at week six and who have a clinical response to the standard infliximab dose should continue on it during the maintenance phase. Still, those who fail to respond (primary non-responders) should switch the drug. Individuals with subtherapeutic ITL should instead be assessed based on their ATI levels. If ATI is undetectable or present at low levels, intensifying therapy is recommended. In contrast, switching therapy should be considered if a high titer of ATI is present [11,14]. In the case of Lisa Tracker, the cutoff is above 200 ng/mL [32].

Unlike previously reported data [8,24,25,26,27], we could not demonstrate a statistically significant exposure–response relationship between infliximab serum concentration quartiles at week six and clinical and endoscopic remission at six months. Still, it must be pointed out that intensifying patients in subtherapeutic ITL and undetectable ATI at week six may have allowed them to improve outcomes at the maintenance phase, reaching remission rates close to those seen in patients with optimal ITL at week six. 

Our study had some limitations: firstly, the lack of fecal calprotectin analysis, which is part of the goals in the IBD treatment and could be assessed as an intermediated surrogate marker of inflammation [15]; secondly, the lack of activity index data at the end of induction did not allow us to assess whether early pTDM prevents primary non-response; thirdly, having a control group to detect differences with conventional (non-TDM-based) management would also have been helpful. Finally, the sample size was limited, and the influence of pTDM in outcomes at one year was not evaluated because the drug was switched from IFX originator to biosimilar by the Brazilian National Public Health system. Furthermore, in the present study, we performed IFX dose intensification for all patients with ITL below 10 µg/mL, but more recent studies have suggested a cutoff of 15 µg/mL at week six [19]. Therefore, some patients might benefit from higher drug levels than those prescribed in this study. Moreover, whether we can use a specific concentration threshold in all patients is still unclear. It is certainly possible that some patients require a different target level due to the difference and intensity in the pathway that drives the inflammatory process. A randomized, controlled, multicenter study (the OPTIMIZE trial) designed to compare the efficacy and safety of two different strategies (pTDM combined with a PK model versus standard of care) in CD patients initiating IFX therapy is currently underway, and we expect this trial to shed further light on this issue [33]. Strengths of our study include the prospective nature of the data, the high rate of follow-up, and the high adherence to the algorithm. Another important point is that the same assay was used to detect drug levels and antibodies in all patients evaluated. 

Therapeutic drug monitoring is a big step toward personalized medicine and optimizing care for patients with IBD. Further prospective studies in large populations, including different subgroups, will help define the role of pTDM at induction, establish an optimal cutoff point, and determine the best time to measure ITL. Steady-state concentration should not be the only indicator for optimizing treatment in these patients, but it is an additional tool to improve response rates and keeping on the drug [7]. Individualized dose regimens will likely be driven by machine learning algorithms based on pharmacogenomics, individualized pharmacokinetic models, and clinical drug responses. 

## Figures and Tables

**Figure 1 biomedicines-11-01757-f001:**
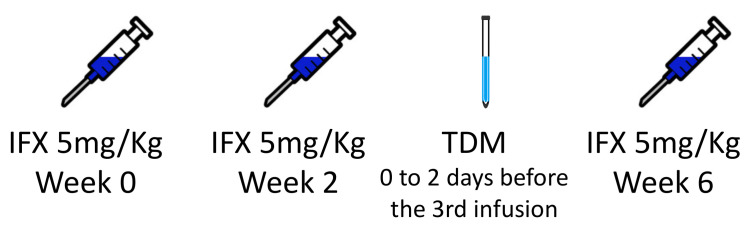
Scheme demonstrating the period for measuring infliximab trough levels and antibodies to infliximab titers. IFX: infliximab; TDM: therapeutic drug monitoring.

**Figure 2 biomedicines-11-01757-f002:**
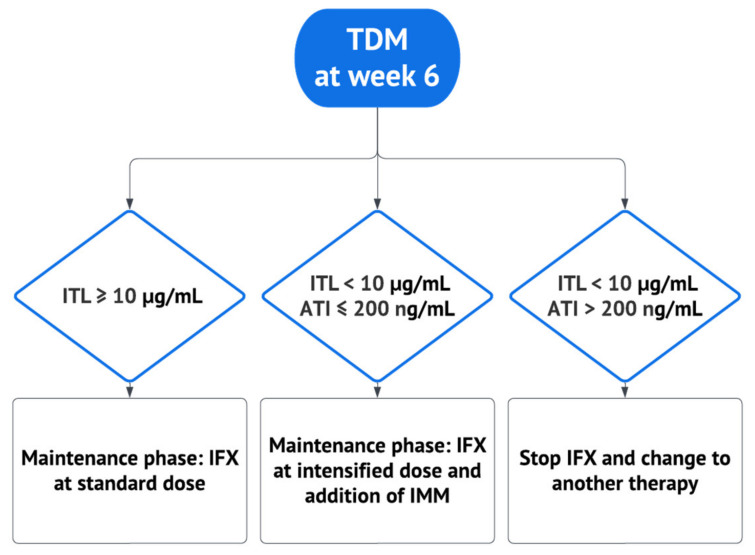
Algorithm guiding therapeutic decisions based on infliximab trough level and antibodies to infliximab levels. ATI: antibodies to infliximab; IFX: infliximab; IMM: immunomodulator; ITL: infliximab trough level; TDM: therapeutic drug monitoring. Adapted from Papamichael et al. [14] and Sparrow et al. [11].

**Figure 3 biomedicines-11-01757-f003:**
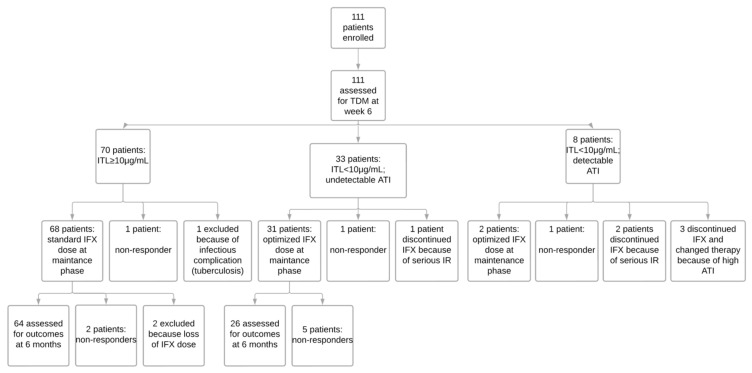
Trial profile. ATI: antibodies to infliximab; IFX: infliximab; IR: infusion reaction; ITL: infliximab trough level; TDM: therapeutic drug monitoring.

**Figure 4 biomedicines-11-01757-f004:**
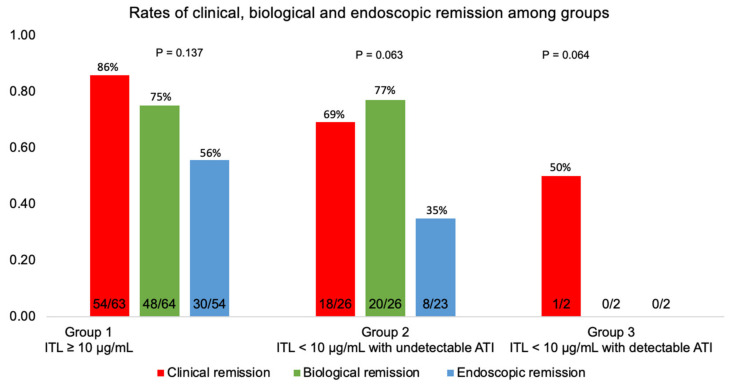
The proportion of patients in clinical, biological, and endoscopic remission among groups classified according to TDM at week six. Group 1: Infliximab trough level (ITL) ≥ 10 µg/mL; group 2: ITL < 10 µg/mL with undetectable antibodies to infliximab (ATI); group 3: ITL < 10 µg/mL with detectable antibodies to infliximab (ATI).

**Figure 5 biomedicines-11-01757-f005:**
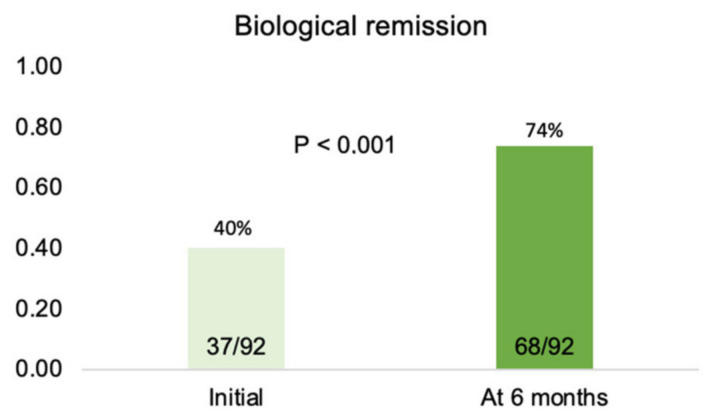
Percentage of patients in biological remission (CRP ≤ 5 mg/L) pre-treatment and after six months of therapy.

**Figure 6 biomedicines-11-01757-f006:**
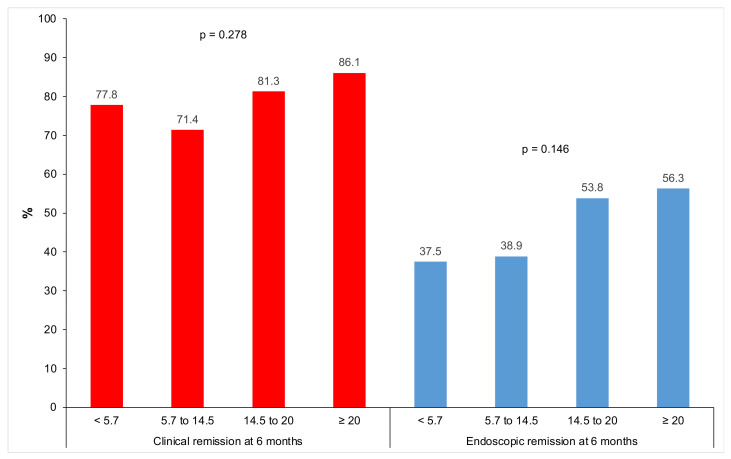
Clinical and endoscopic remission at six months according to infliximab trough level quartiles at week 6.

**Figure 7 biomedicines-11-01757-f007:**
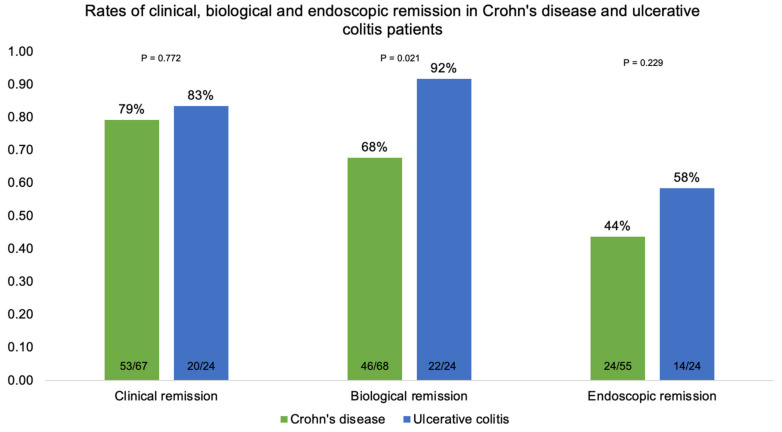
Clinical, biological, and endoscopic remission in patients with Crohn’s disease and ulcerative colitis at six months.

**Table 1 biomedicines-11-01757-t001:** Baseline characteristics of the studied patients (*n* = 111).

Characteristc	Description
Gender, n (%)	
Female	65 (58.6)
Male	46 (41.4)
Age (years), median (IQR)	37 (25; 50)
Body mass index (kg m^−2^), median (IQR)	23.5 (20.5; 26.9)
Current smoker, *n* (%)	10 (9)
IBD type, *n* (%)	
CD	76 (68.5)
UC	35 (31.5)
CD Location, *n* (%)	
L1 (ileal)	26 (34.2)
L2 (colonic)	20 (26.3)
L3 (ileocolonic)	29 (38.2)
L4 (upper GI disease)	1 (1.3)
CD behaviour, *n* (%)	
B1 (nonstricturing, nonpenetrating)	39 (51.3)
B2 (stricturing)	25 (32.9)
B3 (penetrating),	12 (15.8)
Perianal fistulizing disease, *n* (%)	27 (35.5)
UC extent, *n* (%)	
E2 (left-side colitis)	8 (22.9)
E3 (pancolitis)	27 (77.1)
Previous segmental ressection, *n* (%)	16 (14.4)
Right ileocolectomy	8 (7.2)
Disease duration (years), median (IQR)	4 (2;8)
Age at diagnosis (y), median (IQR)	30 (22; 43)
Prior use of advanced therapies for IBD, *n* (%)	
Bio naïve	87 (78.4)
Exposed to 1 biologic	21 (18.9)
Exposed to 2 biologics	3 (2.7)
Exposed to 1 small molecule	2 (1.8)
Concomitant IMM at start of infliximab, *n* (%)	
Aazathioprine	66 (59.5)
6-mercaptopurine	1 (0.9)
Methotrexate	6 (5.4)
Concomitant use of corticosteroids, *n* (%)	25 (22.5)
Initial CRP (mg L^−1^), median (IQR)	7.6 (5; 19)
HBI, median (IQR)	8 (5;12)
PMS, median (IQR)	7 (6; 8)
SES-CD, median (IQR)	13 (8; 17)
Rutgeerts score, median (IQR)	2 (2; 3)
EMS, median (IQR)	3 (2; 3)

A patient may have had more than one surgery. CD: Crohn’s disease; CRP: C-reactive protein; HBI: Harvey-Bradshaw index; IBD: inflammatory bowel disease; IMM: immunomodulator; IQR: interquartile range; MES: Mayo endoscopic score; PMS: partial Mayo score; SES-CD: simple endoscopic score for Crohn’s disease; UC: ulcerative colitis.

**Table 2 biomedicines-11-01757-t002:** Clinical remission rates according to patients’ characteristics and results of bivariate analyses.

Characteristic	Clinical Remission		OR	CI (95%)		*p*
	No	Yes		Inferior	Superior	
Gender, *n* (%)						0.629 ^&^
Male	7 (17.5)	33 (82.5)	1.00			
Female	11 (21.6)	40 (78.4)	0.77	0.27	2.21	
Age (years)			0.99	0.96	1.03	0.631 **
Median (IQR)	36 (26.5; 48.8)	35 (24; 48)				
BMI (kg m^−2^)			0.96	0.86	1.06	0.370 **
Median (IQR)	24.1 (21; 28.2)	23.4 (20.5; 26.8)				
IBD, *n* (%)						0.772 *
CD	14 (20.9)	53 (79.1)	1.00			
UC	4 (16.7)	20 (83.3)	1.32	0.39	4.49	
Previous abdominal surgery, *n* (%)						0.166 *
No	13 (17.1)	63 (82.9)	1.00			
Yes	5 (33.3)	10 (66.7)	0.41	0.12	1.41	
Previous use of adalimumab, *n* (%)						0.079 *
No	12 (16)	63 (84)	1.00			
Yes	6 (37.5)	10 (62.5)	0.32	0.10	1.04	
Corticosteroid - dependence, *n* (%)						0.051 ^&^
No	6 (12.2)	43 (87.8)	1.00			
Yes	12 (28.6)	30 (71.4)	0.35	0.12	1.03	
Duration of disease (years)			0.97	0.90	1.03	0.406 ^£^
Median (IQR)	4.3 (2.4; 11.8)	4 (2; 8)				
Age at diagnosis (years)			1.00	0.96	1.05	0.940 ^£^
Median (IQR)	28 (23.3; 35.5)	29 (20.5; 41.5)				
ITL at week 6 (μg mL^−1^)			1.05	0.98	1.13	0.159 ^£^
Median (IQR)	10.1 (6.7; 20)	18.1 (9; 20)				
Initial CRP (mg L^−1^)			0.99	0.96	1.02	0.427 ^£^
Median (IQR)	9.3 (5; 26.3)	7.3 (5; 19.5)				

^&^ Chi-square test; * Fisher’s exact test; ** Student’s *t*-test; ^£^ Mann–Whitney test. BMI: body mass index; CD: Crohn’s disease; CRP: C-reactive protein; IBD: inflammatory bowel disease; IQR: interquartile range; ITL: infliximab trough level; UC: ulcerative colitis.

**Table 3 biomedicines-11-01757-t003:** Endoscopic remission and association with variables.

Characteristic	Endoscopic Remission		OR	CI (95%)		*p*
	No	Yes		Inferior	Superior	
Gender, *n* (%)						0.607 *
Male	16 (48.5)	17 (51.5)	1.00			
Female	25 (54.3)	21 (45.7)	0.79	0.32	1.94	
Age (years)			1.01	0.98	1.04	0.689 **
Median (IQR)	35 (24.5; 45.5)	34.5 (24; 49)				
BMI (kg m^−2^)			1.01	0.92	1.10	0.855 **
Median (IQR)	23.3 (20.5; 26)	24.3 (20.9; 27)				
IBD, *n* (%)						0.229 *
CD	31 (56.4)	24 (43.6)	1.00			
UC	10 (41.7)	14 (58.3)	1.81	0.69	4.77	
Previous abdominal surgery, *n* (%)						0.878 *
No	34 (51.5)	32 (48.5)	1.00			
Yes	7 (53.8)	6 (46.2)	0.91	0.28	3.00	
Previous use of adalimumab, *n* (%)						0.171 *
No	32 (48.5)	34 (51.5)	1.00			
Yes	9 (69.2)	4 (30.8)	0.42	0.12	1.49	
Corticosteroid-dependence, *n* (%)						0.576 *
No	19 (48.7)	20 (51.3)	1.00			
Yes	22 (55)	18 (45)	0.78	0.32	1.88	
Duration of disease (years)			0.99	0.93	1.06	0.334 ^£^
Median (IQR)	5 (2; 8)	3 (1.9; 9)				
Age at diagnosis (years)			1.02	0.98	1.05	0.438 ^£^
Median (IQR)	25 (20; 33.5)	26 (22.5; 41.3)				
ITL at week 6 (μg mL^−1^)			1.06	0.99	1.13	0.110 ^£^
Median (IQR)	13.7 (5.3; 20)	19.5 (10.9; 20)				
Initial CRP (mg L^−1^)			0.99	0.97	1.02	0.770 ^£^
Median (IQR)	7.5 (5; 26.5)	9.5 (5; 18.3)				

* Chi-square test; ** Student’s *t*-test; ^£^ Mann–Whitney test. BMI: body mass index; CD: Crohn’s disease; CRP: C-reactive protein; IBD: inflammatory bowel disease; IQR: interquartile range; ITL: infliximab trough level; UC: ulcerative colitis.

**Table 4 biomedicines-11-01757-t004:** Biological remission rates according to patients’ characteristics and results of bivariate analyses.

Characteristic	Biological Remission		OR	CI (95%)		*p*
	No	Yes		Inferior	Superior	
Gender, *n* (%)						0.533 ^&^
Male	12 (29.3)	29 (70.7)	1.00			
Female	12 (23.5)	39 (76.5)	1.35	0.53	3.42	
Age (years)			1.00	0.97	1.03	0.780 **
Median (IQR)	32.5 (24; 49.3)	37 (25; 48)				
BMI (kg m^−2^)			0.98	0.89	1.08	0.672 **
Median (IQR)	25 (20.1; 20.6)	23.2 (20.8; 26.9)				
IBD, *n* (%)						0.021 ^&^
CD	22 (32.4)	46 (67.6)	1.00			
UC	2 (8.3)	22 (91.7)	5.26	1.14	24.40	
Previous abdominal surgery, *n* (%)						0.548 *
No	21 (27.6)	55 (72.4)	1.00			
Yes	3 (18.8)	13 (81.3)	1.66	0.43	6.40	
Previous use of adalimumab, *n* (%)						0.134 *
No	17 (22.7)	58 (77.3)	1.00			
Yes	7 (41.2)	10 (58.8)	0.42	0.14	1.27	
Corticosteroid - dependence, *n* (%)						0.983 ^&^
No	13 (26)	37 (74)	1.00			
Yes	11 (26.2)	31 (73.8)	0.99	0.39	2.52	
Duration of disease (years)			0.95	0.89	1.01	0.079 ^£^
Median (IQR)	6 (3; 10.5)	3 (2; 8)				
Age at diagnosis (years)			1.01	0.97	1.05	0.465 ^£^
Median (IQR)	27 (20; 39.3)	29 (23; 41)				
ITL at week 6 (μg mL^−1^)			1.04	0.97	1.11	0.112 ^£^
Median (IQR)	14.1 (9; 19.5)	18.7 (8.8; 20)				
Initial CRP (mg L^−1^)			0.98	0.96	1.00	0.002 ^£^
Median (IQR)	12.5 (7.4; 28.2)	5 (5; 18)				

^&^ Chi-square test; * Fisher’s exact test; ** Student’s *t*-test; ^£^ Mann–Whitney test. BMI: body mass index; CD: Crohn’s disease; CRP: C-reactive protein; IBD: inflammatory bowel disease; IQR: interquartile range; ITL: infliximab trough level; UC: ulcerative colitis.

**Table 5 biomedicines-11-01757-t005:** Final models (multiple logistic regression) adjusted to explain clinical remission.

Outcome	Variables	OR	CI (95%)		*p*
			Inferior	Superior	
Clinical remission at 6 months	Previous use of adalimumab	0.24	0.06	0.97	0.045
	ITL at week 6 (μg mL^−1^)	1.03	0.94	1.12	0.556

ITL: infliximab trough level.

## Data Availability

The data presented in this study are available on request from the corresponding author. The data are not publicly available due to ethics concerns.

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
