# Peer review of "Efficacy of Early Optimization of Infliximab Guided by Therapeutic Drug Monitoring during Induction—A Prospective Trial"

_biomedicines, 2023, doi:10.3390/biomedicines11061757_

Round 1

Reviewer 1 Report

The topic of this manuscript is interesting and fits well the scope of Biomedicines. The reviewer feels it can be accepted after various amendments.

1) The approval number of this clinical trial should be stated clearly.

2) The limitation of this study is obvious and the sample size is not very big.

3) The authors should take notes that score data cannot be analyzed by parametric test. Only non-parametric method should be applied.

4) The bioanalytical method validation information should be included in the manuscript. 

Author Response

June 4th, 2023

Professor Dr. Shaker A. Mousa

Editor-in-chief, Biomedicines

Dear Professor Shaker A. Mousa,

Thank you for the opportunity to revise our manuscript for further consideration for publication in Biomedicines Journal. Below, we provide a point-by-point response to the questions raised by the reviewers. We think the new version of our manuscript incorporating the reviewers’ feedback, comments, and suggestions may improve the wording and emphasize the key messages of the text. We hope you can find it suitable for publication.

Sincerely,

Karoline Soares Garcia, on behalf of all co-authors.

REVIEWER COMMENTS

Reviewer: A

COMMENTS TO THE AUTHOR(S)

The approval number of this clinical trial should be stated clearly.

Thank you for this observation. We added this information in the Ethical Considerations session: “The local Ethical Research Committee approved the study, under number 51767421.0.0000.0068.”

The limitation of this study is obvious and the sample size is not very big.

We agree with the reviewer's comment. However, there is an explanation for the limited sample size. The trial was conducted with patients from the Brazilian Public Health system, and, unfortunately, the supply of the infliximab originator was changed to the biosimilar, which caused the study to be stopped early due to the change in the drug. This fact limited the evaluation of outcomes at one year and the inclusion of new participants after that time.

We included the sample size as a limitation in the discussion session as follows: “Finally, the sample size was limited, and the influence of pTDM in outcomes at one year was not evaluated because the drug was switched from IFX originator to biosimilar by the Brazilian National Public Health system.”

The authors should take notes that score data cannot be analyzed by parametric test. Only non-parametric method should be applied.

Thank you for this feedback. Please allow me to clarify this issue. Parametric tests (Student t-test and logistic regression) were used to analyze age, BMI, and variables related to clinical remission. In contrast, non-parametric tests (Mann-Whitney, Chi-square, and Fisher's exact tests) were used for the variables gender, type of IBD, previous therapies, corticosteroid dependence, age at IBD diagnosis, duration of disease, ITL at week six and serum level of CRP. Each variable is associated with a symbol in the last column in the tables, which indicates the test used, as described at the bottom.

The bioanalytical method validation information should be included in the manuscript.

Thank you for this valuable insight. We have included a sentence exploring Lisa Tracker ®. “Serum ITL and ATI were determined using a commercially available and validated enzyme-linked immunosorbent assay (ELISA) kit (Lisa Tracker ®).  The kit is based on a two-step test with biotinylated anti-human IgG antibodies and horseradish peroxidase streptavidin. The limits of quantification for infliximab were from 0.3 to 20 μg/mL, and for ATI, were from 10 to 200 ng/mL.”

Reviewer 2 Report

The authors evaluated the previously reported algorithm of infliximab guided by TDM. The content is poor. For readers, it is important to inform in which patients the algorithm is appropriate. Evaluation per populations should be performed.

Minor

1.       Figure 3: The word is too small, and it cannot be read.

2.       Figure 4: Which groups were compared? It is recommended that the outcomes are described in Table.

Several sentences are wrong grammar. English language editing is needed.

Author Response

June 4th, 2023

Professor Dr. Shaker A. Mousa

Editor-in-chief, Biomedicines

Dear Professor Shaker A. Mousa,

Thank you for the opportunity to revise our manuscript for further consideration for publication in Biomedicines Journal. Below, we provide a point-by-point response to the questions raised by the reviewers. We think the new version of our manuscript incorporating the reviewers’ feedback, comments, and suggestions may improve the wording and emphasize the key messages of the text. We hope you can find it suitable for publication.

Sincerely,

Karoline Soares Garcia, on behalf of all co-authors.

Reviewer: B

COMMENTS TO THE AUTHOR(S)

The authors evaluated the previously reported algorithm of infliximab guided by TDM. The content is poor. For readers, it is important to inform in which patients the algorithm is appropriate. Evaluation per populations should be performed.

Thank you for this critical feedback. We have included references supporting the utilization of the algorithm during induction in IBD patients, especially when there is an increased risk of higher clearance of the drug. We recognize this could clarify this content. We thus included references from the proposed algorithm by IBD experts. “The algorithm was proposed by Sparrow et al. (11), and Papamichael et al. (14), and it resulted from studies supporting that higher anti-TNF drug levels during induction were associated with favorable outcomes in short and long terms in IBD patients initiating treatment with infliximab, adalimumab or certolizumab pegol. Early optimization of anti-TNF therapy based on this approach might also prevent undesired outcomes, such as primary non-response and the need for surgeries and hospitalizations, especially when there is an increased risk of higher drug clearance. (11,14).”

Analyzes per population were also performed and included as figure 7.

Figure 3: The word is too small, and it cannot be read.

Thank you for highlighting this. We have adjusted the image for better quality.

Figure 4: Which groups were compared? It is recommended that the outcomes are described in Table.

            Thanks for this observation. We have added the group identifications in the table.

Several sentences are wrong grammar. English language editing is needed.

We thank you for your constructive examination of our manuscript. As a result, we are sending the new version of the document revised by a professional English editor.

Round 2

Reviewer 2 Report

The revises is appropriate.